# Analysis of Preferential Flow in Artificial Substrates with *Sedum* Roots for Green Roofs: Experiments and Modeling

**Xuan Chen** [1,2], **Ruifen Liu** [1,*], **Defu Liu** [1,3] **and Xiaokang Xin** [4]

1. Hubei Key Laboratory of Ecological Remediation of River-Lakes and Algal Utilization, School of Civil Engineering, Architecture and Environment at Hubei University of Technology, Wuhan 430068, China
2. Hubei YongYeHang Appraisal Consulting Co., Ltd., Wuhan 430061, China
3. Engineering Research Center of Eco-Environment in Three Gorges Reservoir Region, Ministry of Education, China Three Gorges University, Yichang 443002, China
4. Changjiang Water Resources Protection Institute, Wuhan 430051, China
* Correspondence: ruifen1986@aliyun.com; Tel.: +86-186-9612-6075

**Abstract:** The occurrence of preferential flow in vegetated artificial substrates can weaken the stormwater management performance of green roofs. To explore preferential flow, various plant–substrate combinations that involved two *Sedum* species (*Sedum sarmentosum* and *Sedum lineare*) and two artificial substrates for three depths of 6, 10, and 14 cm were established. Artificial substrates without plants were either perlite-based (namely, PAS) or vermiculite-based (namely, VAS), and they were also set as controls. Thereafter, solute breakthrough experiments were conducted, followed by inverse and forward modeling in Hydrus-1D. Skewness coefficients of all solute breakthrough curves were non-zero, suggesting a prevalence of preferential flow. The Nash–Sutcliffe efficiency coefficients during calibration and validation were greater than 0.7. The obtained hydraulic parameters were different among various vegetated PAS and pure PAS without plants, but appeared the same for the VAS case. Rainfall intensity, plant species, and substrate depth, and the interaction of plant species and substrate depth all had significant effects on PAS preferential flow outflow and index (PFI). Substrate depth had a significant effect on VAS preferential flow and PFI. Since a 10 cm-PAS with *S. lineare* had the smallest PFI of 43.16% in simulation scenarios, its use may better control preferential flow in green roofs.

**Keywords:** green roofs; preferential flow; artificial substrate; *Sedum* roots; solute breakthrough experiments; HYDRUS-1D

## 1. Introduction

Urban stormwater issues, such as inland flooding and water pollution [1], remain challenging in many Chinese cities, and "Sponge City" is a Chinese concept to tackle these issues [2]. Green roof, especially that which is categorized as "extensive green roof" [3] with flat or gentle slope [4], has become one of the important stormwater control measures for sponge city construction. A green roof usually consists of multiple function layers, among which the vegetation layer and substrate layers play important roles in retaining rainwater and detaining runoff [5,6]. When rainwater falls on a green roof, the plant leaves, stems, and branches intercept rainwater, and the substrate layer stores rainwater in its pore structure. Due to the limited depth of the substrate layer, there is a maximum amount of water that substrate can hold within its structure against the pull of gravity [7] (that is, water-holding capacity (WHC) [6] or maximum water capacity [8]). Normally, it is believed that green roof runoff will not occur until the rainwater stored in the substrate pores exceeds the WHC of the substrate [6]. The retained rainwater refers to the difference between rainwater and runoff, and green roofs can be effective in reducing rainwater volume [3]. Detention refers to the temporal delay that occurs between rainwater that is not retained and green roof runoff, and this process can determine the timing and magnitude of peak runoff [9].

*Sedum* species, which are extremely resistant to the harsh conditions of summer drought and winter cold on roofs with stable coverage [10], have been widely used in green roof projects [11–13]. According to the Guidelines for the Planning, Execution, and Upkeep of Green-Roof Sites (hereinafter referred to as FLL standards) [7], an artificial substrate consisting of 80% to 90% *v/v* lightweight aggregate and 10% to 20% *v/v* organic matter favors plant growth, with a quality of nutrient-rich, lightweight, and good permeability. Artificial substrate has now become the preferred choice for green roofs, compared to garden soil and improved soil [14].

Previous green roof studies have showed that plant selection, as well as substrate type and depth, influenced stormwater management in green roofs [15–18]. It should be noted that there is a link between plant root traits and the stormwater management performance of green roofs. MacIvor and Lundholm [19] monitored the hydrological performance of 15 green roofs, each with a monoculture of different plants, but the same substrate consisting of potting soil, brick, perlite, sand, peat, and compost, in the Atlantic Canada coastal region. The monitoring results indicated that the greater the plant root density, the less the rainwater retained. Hu et al. [20] conducted continuous hydrological monitoring on four green roofs with a monoculture of different plants (*Callisia repens*, *Portulaca grandiflora*, *Plectranthus prostratus*, and *Sedum lineare*), but the same substrate consisting of peat soil, perlite, and vermiculite, in Shenzhen, China. It was noted that the larger the diameter of individual roots, the less the rainfall retained. The above facts clearly show that plant roots influence the green roof hydrological performance, and runoff differences among substrates with different plants may link to root-induced changes in pore structures and hydraulic properties of substrates [21]. However, quantitative studies on plant root traits and hydraulic properties of vegetated substrates for green roofs are relatively rare.

Quite often, the rainwater retention effect of green roofs decreases with the increasing amount of rainfall [22]. This can be explained by the limited WHCs of green roofs and may also be associated with the preferential flow in the root-induced macropore channels during large rainfall events [22]. Preferential flow is a non-uniform, non-equilibrium flow [23], a common form of water movement and solute transport [24,25]. With large pores as the preferred paths, the occurrence of preferential flow can cause a rapid transport of water and solute and insufficient contact between substrate and water; as a result, substrate can generate runoff before it reaches its WHC [21,26]. The generation of preferential flow will make green roofs less capable of retaining rainwater and detaining runoff [27], especially for large rainfall events that are critical for urban drainage and flood control [28]. Moreover, in consideration of the interaction between water, heat, and solute [29,30], preferential flow will also influence green roofs' other performances, such as cooling effect and runoff quality improvement.

The generation of preferential flow in green roofs can be the result of a combination of water conditions and internal factors [31]. Water conditions referring to initial water content, rainfall intensity, etc. may affect the time of runoff occurrence and volume [32,33]. Internal factors are mainly characteristics related to the vegetation layer and substrate layer. An artificial substrate compliant with the FLL standards will contain a considerable amount of large particles (>2 mm) and have a limited portion of fine particles (<63 μm). This composition would create numerous large pores (e.g., 0.03–3.00 mm) that are likely to cause preferential flow to occur [34]. Liu and Fassman-Beck [35] detected preferential flow in non-vegetated substrate by indoor experiments and simulation methods, indicating the occurrence of preferential flow in substrate with porous structures at low water content. However, this study did not consider the role of plants. Plant roots account for a large proportion of the green roof substrate layer [36], and the pore channels formed by plant root are also one of the important mechanisms for preferential flow generation [37]. Zhang et al. [38] showed that in both the mixture and as a monoculture, an herbaceous plant (*Stypandra glauca*) created preferential flow pathways in green roofs. However, very few studies have provided quantitative data about root traits (e.g., dimeter and volume density) of commonly used *Sedum* species for the stormwater management purpose, and

detailed investigation about preferential flow in green roof substrates with *Sedum* species remains lacking.

The purposes of this paper are: (1) to detect the occurrence of preferential flow in various plant–substrate combinations by indoor solute breakthrough experiments, (2) to characterize the substrate hydraulic properties of each combination, and (3) to analyze the effects of plant species, substrate depth, rainfall intensity, and initial water content on the preferential flow development in plant–substrate combinations. Two artificial substrates with various plant species and substrate depths were subjected to solute breakthrough experiments to detect the occurrence of preferential flow. The Hydrus-1D model, validated by experimental data, was used to obtain the hydraulic parameters of each combination and to investigate the influence of different factors on preferential flow.

## 2. Materials and Methods

### 2.1. Plants and Substrates

Two *Sedum* species indigenous to China, *Sedum sarmentosum* (*SS*) and *Sedum lineare* (*SL*), and two artificial substrates in accordance with the FLL standards, perlite-based artificial substrate (PAS) and vermiculite-based artificial substrate (VAS), were selected for experiments. The basic physical (e.g., saturated hydraulic conductivity ($K_s$)) and chemical properties of PAS and VAS are shown in Table 1 [39]. For each artificial substrate, both plant species were propagated by stem cuttings in monoculture at three depths (6 cm, 10 cm, and 14 cm) in 10 cm-diameter acrylic cylinders and grown in an artificial climate chest for 103 days to ensure an excellent plant coverage [39]. Thereafter, plant root characteristics, such as root volume density, and $K_s$ of these 12 vegetated substrates of varying depths (Table 2) were measured [39].

**Table 1.** Characteristics of the artificial substrates [mean (SE)].

| Characteristics | Substrate Type | |
|---|---|---|
| | PAS | VAS |
| Components (% by volume) | 90% perlite (<6 mm) 10% chicken manure | 90% vermiculite (<5 mm) 10% chicken manure |
| Bulk density (g/cm$^3$) | 0.21 (0.01) | 0.34 (0.01) |
| Total porosity (%) | 91.40 (0.01) | 78.80 (0.02) |
| WHC (%) | 36.65 (1.33) | 64.05 (1.55) |
| Organic matter content (g/kg) | 31.15 (2.72) | 38.64 (2.60) |
| $K_s$ (cm/min) | 54.45 (0.19) | 18.48 (1.39) |

**Table 2.** Characteristics of plant roots and vegetated artificial substrates [mean (SE)].

| Substrate Depth-Substrate Type-Plant Species | Root Volume Density /(mm$^3$/cm$^3$) | Root Volume Density of 0.2–0.4 mm Roots/(mm$^3$/cm$^3$) | $K_s$ /(cm/min) |
|---|---|---|---|
| 6 cm-PAS-*SS* | 0.63 (0.00) | 0.33 (0.00) | 2.12 (0.17) |
| 10 cm-PAS-*SS* | 0.42 (0.01) | 0.33 (0.01) | 1.97 (0.19) |
| 14 cm-PAS-*SS* | 1.37 (0.02) | 0.20 (0.01) | 0.56 (0.00) |
| 6 cm-PAS-*SL* | 0.71 (0.03) | 0.31 (0.01) | 2.64 (0.09) |
| 10 cm-PAS-*SL* | 0.02 (0.00) | 0.01 (0.00) | 0.68 (0.06) |
| 14 cm-PAS-*SL* | 1.13 (0.02) | 0.15 (0.00) | 0.74 (0.05) |
| 6 cm-VAS-*SS* | 2.46 (0.04) | 0.37 (0.02) | 19.19 (0.54) |
| 10 cm-VAS-*SS* | 0.16 (0.00) | 0.09 (0.00) | 14.14 (1.31) |
| 14 cm-VAS-*SS* | 5.66 (0.06) | 0.14 (0.01) | 12.77 (0.57) |
| 6 cm-VAS-*SL* | 1.59 (0.02) | 0.59 (0.01) | 16.94 (1.28) |
| 10 cm-VAS-*SL* | 1.31 (0.02) | 0.44 (0.01) | 15.44 (0.67) |
| 14 cm-VAS-*SL* | 8.98 (0.24) | 0.15 (0.01) | 13.60 (0.67) |

### 2.2. Solute Breakthrough Experiments

For preferential flow detection in various plant–substrate settings, solute breakthrough experiments were conducted. For each substrate type (PAS or VAS), possible influential factors, such as plant species, substrate depth, rainfall event, and initial water content, were all considered (Figure 1). The level settings of each factor are shown in Figure 1, while plant species, substrate depth, and rainfall event were set as 3-level and initial water content was set as a virtual level, producing a 4-factor, 3-level, orthogonal experimental design for each substrate type. The applied rainfalls of 2 a, 5 a, and 10 a corresponded to 1 h-duration storms for return periods of 2, 5, and 10 years in Wuhan City in central China, with rainfall intensities of 3.8 cm/h, 5.4 cm/h, and 6.6 cm/h, respectively [40]. The initial water content was either water-holding capacity (WHC) or mild drought conditions (MDC) to represent a typical wet/dry moisture condition in green roofs in Wuhan's climate [40]. For each substrate type, nine sets of experiments were conducted, as referring to nine lines in Figure 1. Therefore, there were 18 sets of experiments for the two substrates, and with each set repeating for three times, 54 sets of solute breakthrough experiments were conducted in total.

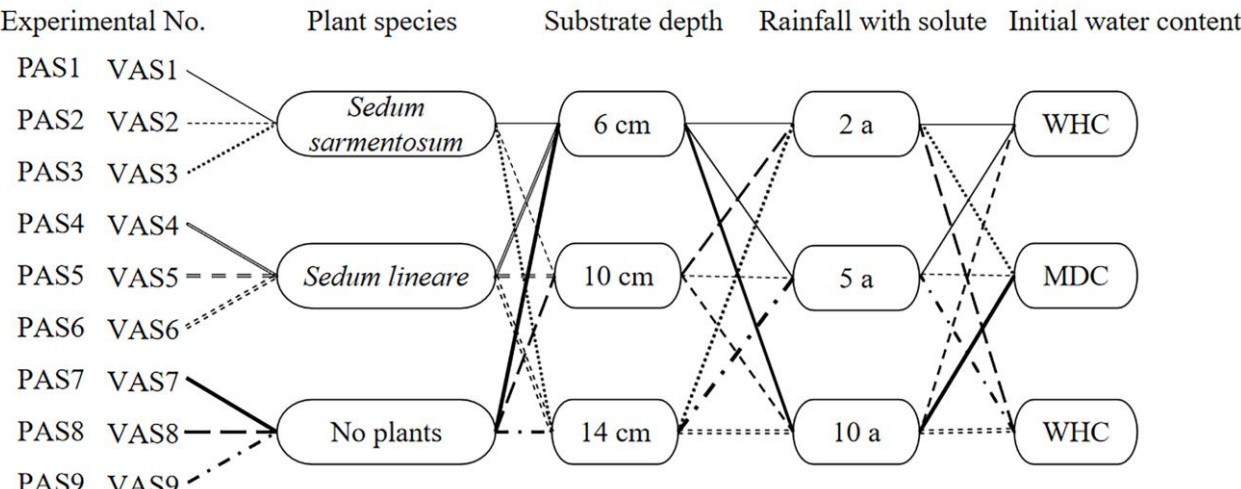

**Figure 1.** Orthogonal design of solute breakthrough experiments.

Figure 2 shows the apparatus for solute breakthrough experiments, which consisted of a Mariotte bottle (volume 4 L), a rainfall device (37 needles of 0.45 mm diameter for inflow), an acrylic column (diameter 10 cm, loading plant–substrate), an outflow collection container, and an automatic weighing scale (capacity 30 kg, accuracy ±0.1 g). NaCl solution of concentration 1 g/L was used as the tracer, which was dosed through the Mariotte bottle. After 60 min of dosing, the NaCl solution was replaced with deionized water without changing intensity [40]. The outflows from the acrylic column were measured automatically by the weighing instrument at 3 min intervals, and outflow water samples were also collected manually for NaCl concentration measurements. Since the $K_s$ values (Tables 1 and 2) of different plant–substrate combinations were notably larger than the applied rainfall intensities, neither ponding nor overflow occurred in all experiments, and the measured outflow was equal to runoff. The cumulative outflow mass [g] was first converted into outflow volume [L] by assuming a water density of 1 g/cm$^3$, and was then converted into water depth [cm] by dividing the column surface area of 10 cm diameter. Taking PAS1 as an example (Figure 1), the cumulative outflow process is shown in Figure 3a. The negative values represent the outflow direction vertical downward. The result of the related solute breakthrough curve is also shown in Figure 3b. The $C$ and $C_0$ [g/L] are the outflow and inflow solute concentration, respectively. $V$ [L] is the cumulative outflow volume with time. $V_0$ [L] is the infiltrated water volume within the substrate pores, equal to the volume of rainfall minus the volume of outflow in the same time.

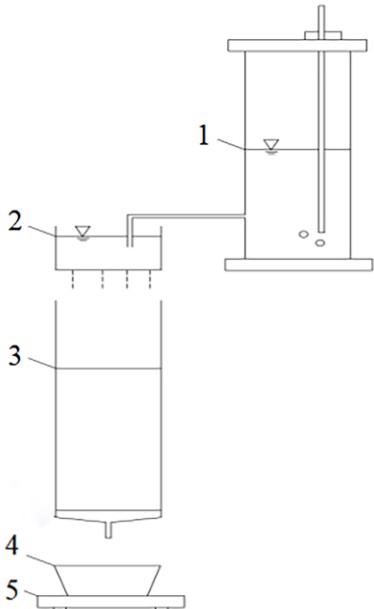

**Figure 2.** Apparatus for solute breakthrough experiments. 1—Mariotte bottle, 2—rainfall device, 3—acrylic column, 4—outflow collection container, 5—automatic weighing scale.

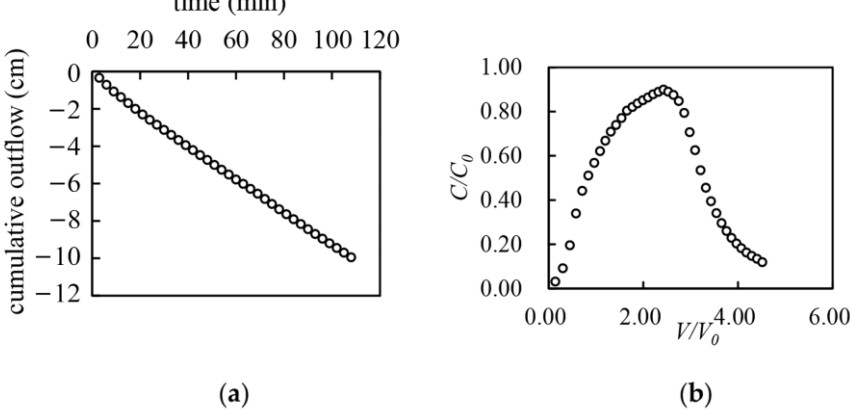

|  (a)  |  (b)  |

**Figure 3.** An example of measured data. (**a**) Cumulative outflow curve; (**b**) Solute breakthrough curve.

### 2.3. Preferential Flow Detection

The shape of the solute breakthrough curve can be quantified by the temporal moments method to determine whether preferential flow is occurring [41]. The temporal moments (*M*) are described as [42]:

$$M_P = \int_0^\infty T^P c(Z,T)/c_0 \, \mathrm{d}T \quad p = 0,1,2,\dots \tag{1}$$

where $p$ is the order of the moments; $T$ is equal to $V/V_0$; $c(Z,T)$ and $c_0$ are the time-dependent outflow and the initial solute concentrations [g/L], respectively; and $Z$ is the dimensionless spatial coordinate.

The temporal moments method also uses the standard moments ($\mu'_p$) and central moments ($\mu_P$), as defined by the following equations:

$$\mu'_p = M_P/M_0 \tag{2}$$

$$\mu_P = \frac{1}{M_0} \int_0^\infty \left(T - \mu'_1\right)^P c(Z,T)/c_0 \mathrm{d}T \quad p = 0,1,2,\dots \tag{3}$$

The first-order standard moments $(\mu_1')$ describe the breakthrough time of the tracer in the solute breakthrough experiment, and the second-order central moments $(\mu_2)$ describe the degree of dispersion of the solute breakthrough curve. The third-order central moments $(\mu_3)$ are used to quantitatively describe the asymmetry of the solute breakthrough curve. The dimensionless skewness coefficient $(S)$ then can be obtained as below:

$$S = \mu_3 / {\mu_2}^{3/2} \tag{4}$$

If $S < 0$, the solute breakthrough curve is a rightward biased curve, and if $S > 0$, the solute breakthrough curve is a leftward biased curve, and if $S = 0$, the solute breakthrough curve is symmetric [43,44]. When preferential flow occurs, $S \neq 0$.

*2.4. Determination of Substrate Hydraulic Parameters*

2.4.1. Calculation of Substrate Hydraulic Parameters

Substrate hydraulic parameters were calculated using inverse modeling by Hydrus-1D [45]. In Hydrus-1D, the water flow module and the inversion module were activated, in which the cumulative outflow data (Figure 3a) were set as the objective function, and the nonlinear Levenberg–Marquardt algorithm was used to minimize the objective function [46]. Once preferential flow was detected (Section 2.3), a dual permeability model [47,48] was selected to describe water flow movement. The dual-permeability model (Equations (5)–(9)) assumes that the porous substrate consists of two interacting, overlapping pore domains. The micropores with relatively low permeability are the matrix domain (subscript *m*), and the high-permeability preferential flow paths, such as large pores and fractures between the matrix, are the preferential flow domain (subscript *f*). Both domains are quantified separately using the two coupled Darcy–Richard equations (Equations (5) and (6)). In these equations, substrate hydraulic parameters $(\theta_r, \theta_s, a, n, K_s,$ and $l)$ defining water retention curves and hydraulic conductivity functions were needed, and the van Genuchten–Mualem formula [46] was used to fit these parameters.

$$\frac{\partial \theta_f}{\partial t} = \frac{\partial}{\partial z}\left[K_f\left(\frac{\partial h_f}{\partial z} + 1\right)\right] - S_f - \frac{\Gamma_w}{\omega} \tag{5}$$

$$\frac{\partial \theta_m}{\partial t} = \frac{\partial}{\partial z}\left[K_m\left(\frac{\partial h_m}{\partial z} + 1\right)\right] - S_m + \frac{\Gamma_w}{1 - \omega} \tag{6}$$

$$\theta = \omega \theta_f + (1 - \omega)\theta_m \tag{7}$$

$$\Gamma_w = \alpha_w \left(h_f - h_m\right) \tag{8}$$

$$\alpha_w = \frac{\beta}{a^2} \Gamma K_a \tag{9}$$

where $\theta_f$, $\theta_m$, $\theta$ are the water content of the preferential flow domain, matrix domain, and the entire domain, respectively, [cm$^3$·cm$^{-3}$]; $t$ is the simulation time [min]; $z$ is the vertical coordinate positive upward [cm]; $S_f$ and $S_m$ are the plant water uptake rates of the preferential flow domain and matrix domain, respectively [min$^{-1}$]; $K_f$ and $K_m$ are the unsaturated hydraulic conductivity of the preferential flow domain and the matrix domain, respectively [cm·min$^{-1}$]; $h_f$ and $h_m$ are the matric potential of the preferential flow domain and the matrix domain, respectively [kPa]; $\omega$ is the proportion of the preferential flow domain to the whole domain [dimensionless]; $\Gamma_w$ is the water exchange rate between the two domains [min$^{-1}$]; $\alpha_w$ is the first-order mass transfer coefficient for water [cm$^{-1}$·min$^{-1}$]; $\beta$ is a dimensionless geometry-dependent shape factor; $\Gamma$ is a dimensionless scaling factor; $a$ is the distance between the center of the matrix domain and the boundary of the preferential flow domain [cm]; and $K_a$ is the effective hydraulic conductivity of the fracture-matrix interface [cm·min$^{-1}$].

For inverse modeling, substrate geometry values and initial water contents were set according to the orthogonal design of solute breakthrough experiments (Figure 1). The corresponding spatial-temporal discretization settings were given in Chen's previous study [39]. Since constant rainfall intensities were applied in the experiments, the upper boundary was set as the constant flux boundary. According to the apparatus setting (Figure 2), the lower boundary was set as the seepage face. Due to the short duration of rainfall, evaporation and plant uptake were not considered [49], and therefore, $S_f, S_m$ in Equation (5) and Equation (6) were 0. Some parameters, such as $a$, $\beta$, and $\Gamma$ in Equation (9), were determined as 0.1 [33,47], 15 (spherical shape assumption), and 0.4 (empirical value) [50], respectively. Still, there were other parameters needed for Equations (5)–(9), including hydraulic parameters of the matrix domain ($\theta_{rm}$ (taking the value of 0), $\theta_{sm}$, $a_m$, $n_m$, $K_{sm}$, $l_m$ (pore curvature, generally taking the value of 0.5)), hydraulic parameters of the preferential flow domain ($\theta_{rf}$ (taking the value of 0.5), $\theta_{sf}$, $\alpha_f$, $n_f$, $K_{sf}$, $l_f$ (taking the value of 0.5)), the parameter of the interface ($K_a$), and the dimensionless factor ($\omega$). Constraints on those unspecified parameters were given to ensure an overall unique solution and convergence in the parameter optimization [51]. Based on substrate physical properties (Table 2), the constraint of saturated water content (that is, the sum of $\theta_{sm}$ and $\theta_{sf}$) of PAS was set as <0.90, and the constraint of saturated water content of VAS was set as <0.78. Since $\alpha_m, \alpha_f, n_m, n_f$ were related to the physical properties of the particles, and the empirical parameter range was set as $\alpha \in (0.001, 0.01), n \in (2, 5)$ [52]. The constraint of hydraulic conductivities of the two domains was set as $K_{sm} + K_{sf} \leq K_s$. The empirical range of $K_a$ was $10^{-7}$–$10^{-4}$ when preferential flow occurred [47]. Based on the measured and modeled values of the objective function, the coefficient of determination $R^2$ (Equation (10), [52]) and the Nash–Sutcliffe efficiency coefficient NSE (Equation (11), [53]) were calculated to determine the optimal parameters.

$$R^2 = \left[ \frac{\sum_{i=1}^{N}(O_i - \bar{O})(P_i - \bar{P})}{\left[\sum_{i=1}^{N}(O_i - \bar{O})^2\right]^{0.5}\left[\sum_{i=1}^{N}(P_i - \bar{P})^2\right]^{0.5}} \right]^2 \tag{10}$$

$$\text{NSE} = 1 - \frac{\sum_{i=1}^{N}(P_i - O_i)^2}{\sum_{i=1}^{N}(O_i - \bar{O})^2} \tag{11}$$

where $N$ is the total number of observations; $P_i$ and $O_i$ are, respectively, the $i$th modeled and observed values ($i$ = 1, 2, ... , $N$); and $\bar{P}$ and $\bar{O}$ are the mean modeled and observed values, respectively. The $R^2$ values close to 1 indicate that variations of the observed values can be captured well in the modeling. NSE can range from $-\infty$ to 1, with a closer value of 1 representing a more perfect match [52,53].

2.4.2. Validation of Substrate Hydraulic Parameters

Substrate hydraulic parameters obtained from the inverse modeling were validated by the forward modeling for the solute transport process in Hydrus-1D, and the dual permeability model (Equations (5)–(9)) was used for the associated water flow process. The classical convection-dispersion equation to describe the solute transport process based on water transport is as follows [46]:

$$\frac{\partial \theta_f c_f}{\partial t} + \rho \frac{\partial s_f}{\partial t} = \frac{\partial}{\partial z}\left(\theta_f D_f \frac{\partial c_f}{\partial z}\right) - \frac{\partial q_f c_f}{\partial z} - \phi_f - \frac{\Gamma_s}{w} \tag{12}$$

$$\frac{\partial \theta_m c_m}{\partial t} + \rho \frac{\partial s_m}{\partial t} = \frac{\partial}{\partial z}\left(\theta_m D_m \frac{\partial c_m}{\partial z}\right) - \frac{\partial q_m c_m}{\partial z} - \phi_m + \frac{\Gamma_s}{1-w} \tag{13}$$

$$\Gamma_s = \omega_{dp}(1-w)\theta_m\left(c_f - c_m\right) + \Gamma_w c^* \tag{14}$$

where $C_f$, $C_m$ are the concentrations of the preferential flow domain and the matrix domain [g·cm$^{-3}$]; ρ is the bulk density of the substrate [g·cm$^{-3}$]; $D_f$, $D_m$ are the sorbed concentrations of the preferential flow domain and the matrix domain [g·g$^{-1}$]; $q_f$, $q_m$ are the volumetric fluid flux densities of the preferential flow domain and the matrix domain [cm·s$^{-1}$]; $\phi_f$, $\phi_m$ are sink-source terms that account for various zero- and first-order or other reactions in both domains [g·cm$^{-3}$·s$^{-1}$]; $\Gamma_s$ is the solute mass transfer term [g·cm$^{-3}$·min$^{-1}$]; $\omega_{dp}$ is the first-order solute mass transfer coefficient [min$^{-1}$]; and $c^* = c_f$ for $\Gamma_w > 0$ and $c^* = c_m$ for $\Gamma_w < 0$.

Most settings of the water flow process for the forward modeling in Hydrus 1D were the same as those for the inverse modeling, such as spatial-temporal discretization, initial values, and boundary conditions. The additional inputs as required by the solute transport process were set according to the solute breakthrough experiments (Section 2.2). The molecular diffusion coefficient in free water $D_w$ for Cl$^-$ was 1.7 cm$^2$/day, and the dispersion coefficient $D_L$ was 1/10 of the corresponding substrate depth (Figure 1). Substrate bulk densities are given in Table 1. The incoming solute concentration was 1 g/L, and the solute dosing time was 60 min (Section 2.2). The forward modeling predicted solute concentrations at different moments, and based on modeled and observed concentrations, $R^2$ and NSE were calculated to assess the rationality of the substrate hydraulic parameters.

### 2.5. Preferential Flow and Influential Factors

Based on the constructed Hydrus-1D model with validated parameters, four influential factors, including plant species, substrate depth, rainfall intensity, and initial water content, can be varied, according to the control variable method [49] to explore the law of preferential outflow for different plant–substrates. A total of 54 simulated conditions were established [39]. This study focuses on conceptual understanding and describing the flow process rather than performing parameter optimization or stochastic model analysis.

For each simulated condition, the solute breakthrough curve was obtained, and its skewness coefficient was calculated for the preferential flow detection (Section 2.3). In addition, the preferential outflow and the preferential flow index (PFI, the percentage of the preferential outflow to the total water flow [33]) based on simulation results were also obtained. Multi-factor ANOVA was used to test whether the main effects and interaction effects of different influential factors on preferential outflow and PFI were significant. The coefficient of variation, $C_v$ [54], was used to describe the variance of preferential outflow and PFI among simulation conditions [55]. According to Nielsen's classification criteria [56], $C_v \leq 10\%$ indicates a weak coefficient of variation, $10\% < C_v < 100\%$ indicates a medium coefficient of variation, and $C_v \geq 100\%$ indicates a strong coefficient of variation.

## 3. Results and Discussion

### 3.1. Preferential Flow Detection

The characteristics of the solute breakthrough curves corresponding to the 18 experimental sets of the 2 artificial substrates are shown in Table 3. It can be seen that the *S* values of all curves are not 0. Among them, the *S* values of the vegetated and non-vegetated PAS are −0.06–0.37 (PAS1–PAS6) and 0.01–0.21 (PAS7–PAS9), respectively. The *S* values of the vegetated and non-vegetated VAS are 0.01–0.30 (VAS1–VAS6) and 0.17–0.61 (VAS7–VAS9), respectively. The results indicate that preferential flow commonly occurs in the green roof plant–substrate combinations. According to the existing literature [57,58], the occurrence of preferential flow is related to the non-homogeneity of the substrate, plant roots, and moisture conditions, which will be discussed later in Section 3.3.

**Table 3.** Summary of solute breakthrough curve characteristics.

| Experimental No. | $M_0$ | $M_1$ | $\mu_1'$ | $\mu_2$ | $\mu_3$ | $S$ | Experimental No. | $M_0$ | $M_1$ | $\mu_1'$ | $\mu_2$ | $\mu_3$ | $S$ |
|---|---|---|---|---|---|---|---|---|---|---|---|---|---|
| PAS1 | 2.34 | 0.53 | 0.23 | 1.61 | −0.12 | −0.06 | VAS1 | 1.28 | 0.58 | 0.5 | 0.41 | 0.01 | 0.05 |
| PAS2 | 1.18 | 0.48 | 0.41 | 0.47 | 0.09 | 0.27 | VAS2 | 0.79 | 0.49 | 0.62 | 0.22 | 0.01 | 0.06 |
| PAS3 | 0.81 | 0.35 | 0.43 | 0.29 | 0.05 | 0.34 | VAS3 | 0.45 | 0.47 | 1.05 | 0.07 | 0.01 | 0.30 |
| PAS4 | 1.13 | 0.42 | 0.37 | 0.57 | 0.16 | 0.37 | VAS4 | 0.87 | 0.51 | 0.58 | 0.24 | 0.01 | 0.05 |
| PAS5 | 1.26 | 0.48 | 0.38 | 0.57 | 0.16 | 0.37 | VAS5 | 0.94 | 0.52 | 0.55 | 0.26 | 0.03 | 0.23 |
| PAS6 | 1.25 | 0.50 | 0.40 | 0.49 | 0.07 | 0.22 | VAS6 | 0.87 | 0.50 | 0.58 | 0.25 | 0.00 | 0.01 |
| PAS7 | 3.39 | 0.62 | 0.18 | 2.46 | 0.44 | 0.11 | VAS7 | 1.90 | 0.48 | 0.25 | 1.37 | 0.98 | 0.61 |
| PAS8 | 0.77 | 0.40 | 0.52 | 0.31 | 0.00 | 0.01 | VAS8 | 0.62 | 0.39 | 0.63 | 0.18 | 0.01 | 0.17 |
| PAS9 | 0.76 | 0.37 | 0.49 | 0.26 | 0.03 | 0.21 | VAS9 | 0.57 | 0.37 | 0.64 | 0.17 | 0.02 | 0.26 |

### 3.2. Substrate Hydraulic Parameters

#### 3.2.1. Results

Figure 4 show the modeled cumulative outflows from the inverse modeling and the corresponding observed outflows. It can be seen that the calculated $R^2$ (Equation (10)) are in the range of 0.998–0.999, and NSE (Equation (11)) are in the range of 0.741–0.997. Figure 5 shows the predicted outflow concentrations from the forward modeling and the corresponding measured concentrations. It shows that $R^2$ are in the range of 0.937–0.993, and NSE are in the range of 0.741–0.973. These data indicate that the substrate hydraulic parameters (Table 4) obtained from the inverse modeling are validated for the forward modeling and can be further used for the preferential flow simulation of different plant–substrates combinations (Section 3.3). In Table 4, the hydraulic parameters of the matrix domain remain constant for each substrate, irrespective of plant–substrate combinations, as the dual permeability model assumes that the root system only make changes to the preferential flow domain [59]. Considering the significant effect of plant root traits on $K_s$ of PAS [60], the hydraulic parameters of the preferential flow domain of PAS are varied. However, since there was no insignificant difference in $K_s$ of VAS due to the root system [60], hydraulic properties of the preferential flow domain of VAS can be viewed as the same.

**Table 4.** Summary of substrate hydraulic parameters.

| Name | Matrix Domain $\theta_{sm}$/ (cm³·cm⁻³) | $\alpha_m$/ (cm⁻¹) | $n_m$ | $K_{sm}$/ (cm·min⁻¹) | Preferential Flow Domain $\theta_{sf}$/ (cm³·cm⁻³) | $\alpha_f$/ (cm⁻¹) | $n_f$ | $K_{sf}$/ (cm·min⁻¹) | $\omega$ | $K_a$/ (cm·min⁻¹) |
|---|---|---|---|---|---|---|---|---|---|---|
| 6 cm-PAS-*SS* | | | | | 0.33 | 0.050 | 2.1 | 2.0 | 0.12 | |
| 10 cm-PAS-*SS* | | | | | 0.35 | 0.050 | 2.1 | 1.8 | 0.11 | |
| 14 cm-PAS-*SS* | | | | | 0.27 | 0.002 | 1.5 | 0.4 | 0.05 | |
| 6 cm-PAS-*SL* | 0.150 | 0.008 | 2.50 | 0.100 | 0.36 | 0.054 | 2.0 | 2.5 | 0.14 | 0.75 × 10⁻⁶ |
| 10 cm-PAS-*SL* | | | | | 0.30 | 0.005 | 1.8 | 0.5 | 0.07 | |
| 14 cm-PAS-*SL* | | | | | 0.30 | 0.005 | 1.8 | 0.6 | 0.07 | |
| pure PAS | | | | | 0.75 | 0.009 | 3.8 | 54.4 | 0.60 | 0.16 × 10⁻⁶ |
| 6 cm-VAS-*SS* | | | | | | | | 19.1 | | |
| 10 cm-VAS-*SS* | | | | | | | | 14.1 | | |
| 14 cm-VAS-*SS* | | | | | | | | 12.7 | | |
| 6 cm-VAS-*SL* | 0.131 | 0.011 | 2.41 | 0.105 | 0.60 | 0.008 | 2.618 | 16.9 | 0.026 | 0.75 × 10⁻⁶ |
| 10 cm-VAS-*SL* | | | | | | | | 15.4 | | |
| 14 cm-VAS-*SL* | | | | | | | | 13.6 | | |
| pure VAS | | | | | | | | 18.8 | | |

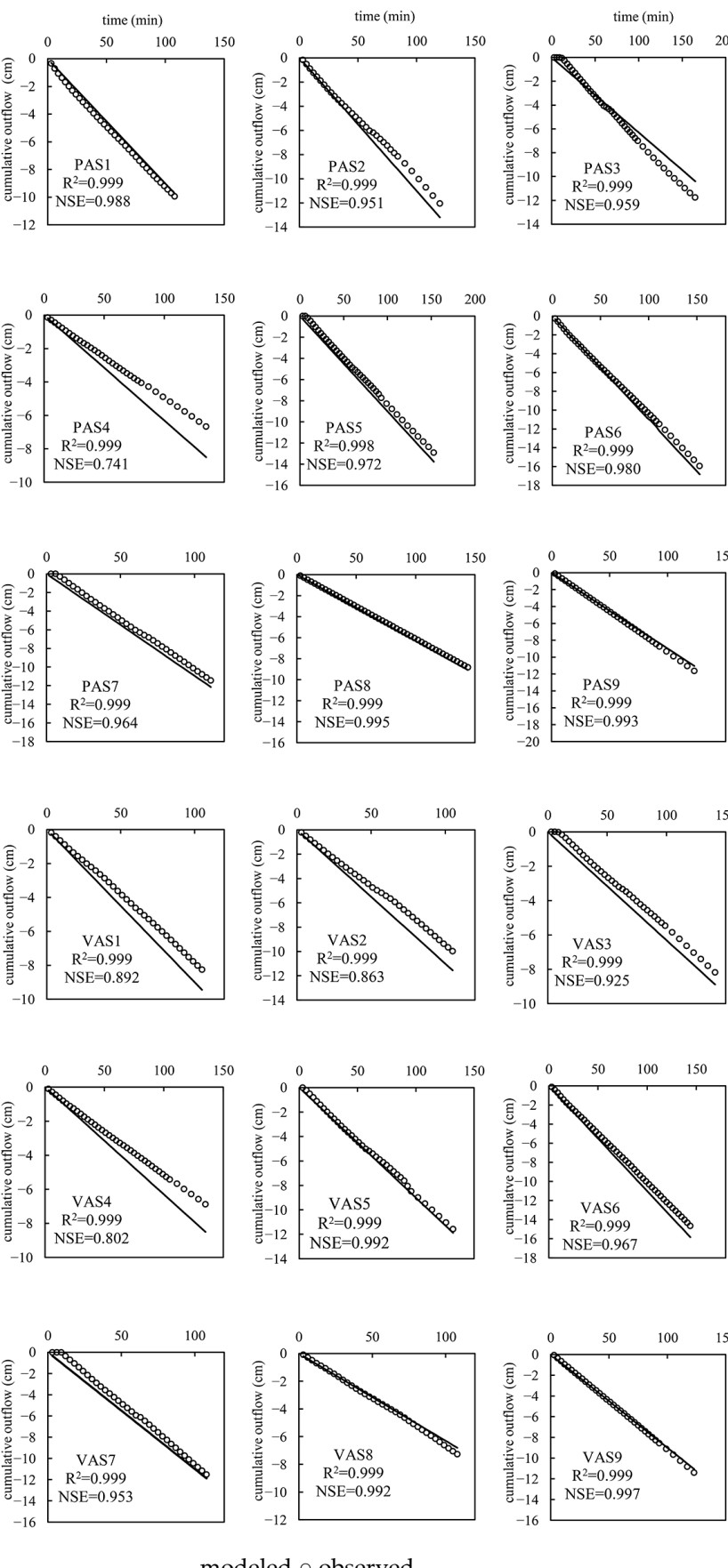

**Figure 4.** Comparison of modeled and observed values of cumulative outflow in inverse modeling.

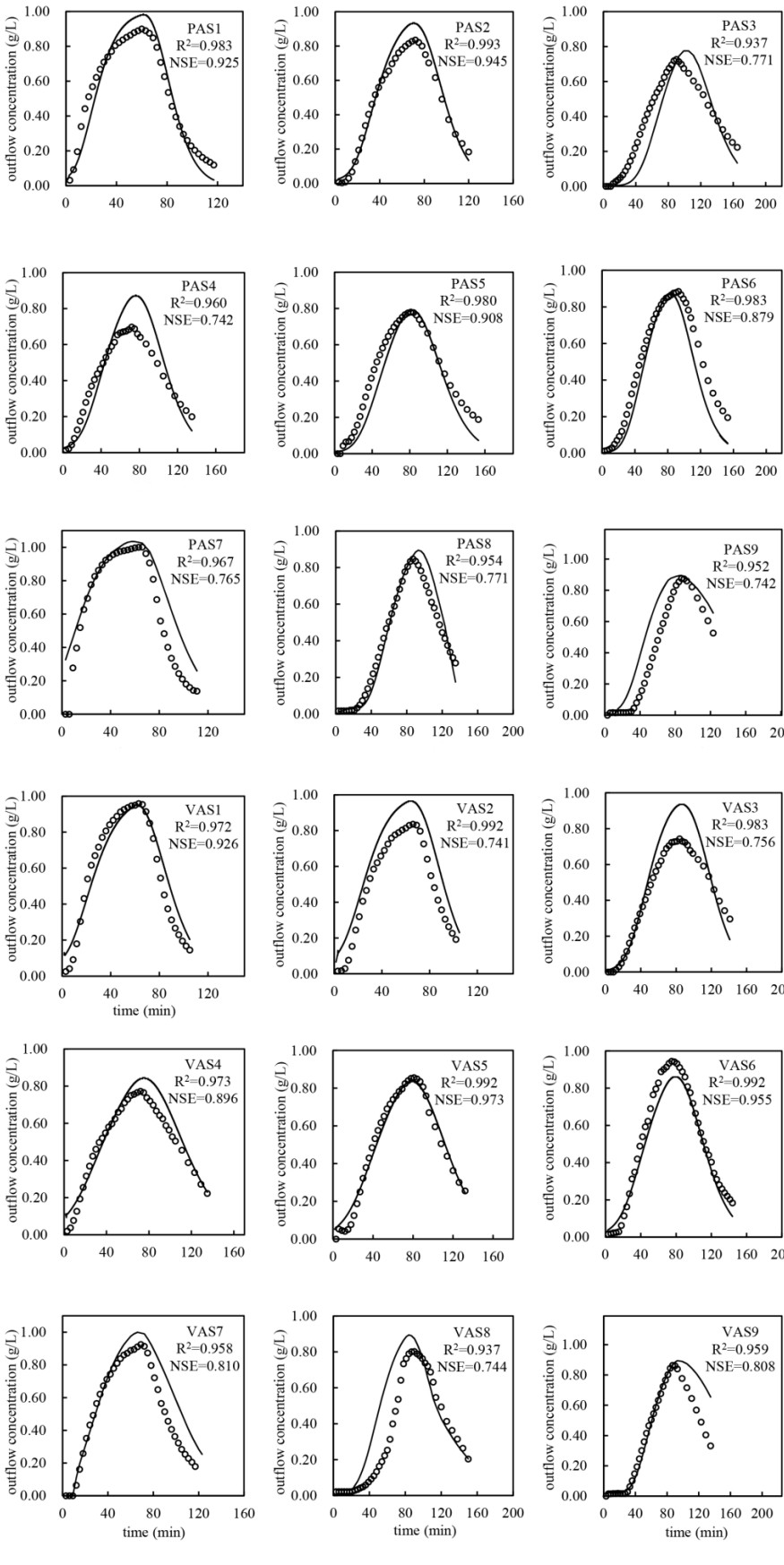

– modeled ○ observed

**Figure 5.** Comparison of predicted and measured outflow concentrations in forward modeling.

### 3.2.2. Implications

Based on the obtained hydraulic parameters (Table 4), the corresponding water retention curves of PAS in preferential flow domain can be plotted. A water retention curve reflects the variation of pore water in the substrate with the matric potential, and also indirectly reflects the distribution of pore size in the substrate [61,62]. It can be seen from Figure 6 that curves from vegetated PAS are significantly different from that from pure PAS. The initial stable water content in high matrix potentials (i.e., around 0 kPa) from pure PAS is noticeably greater than those from the other curves, and afterwards, the decrease in water content, along with lower matrix potentials, is much steeper than those from the other curves. Among these vegetated PAS, curves also show various differences in terms of the initial stable water content, the decreasing slope, and the specific matric potential that the stable water content starts to decrease. Those differences in the water retention curves of PAS indicate different pore structures are present due to different root characteristics, and further exploration, therefore, is made below.

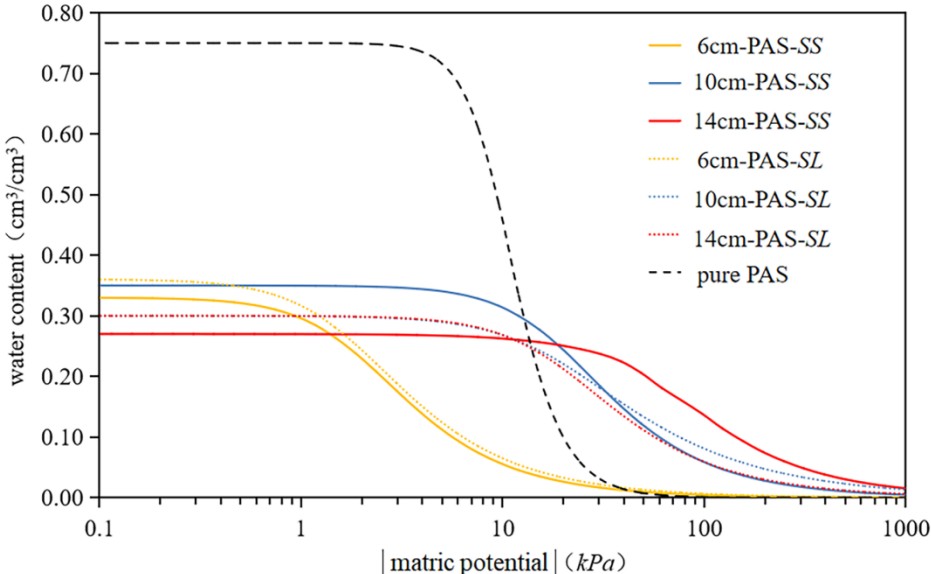

**Figure 6.** Water retention curves of PAS (preferential flow domain).

To further analyze the root-induced changes of hydraulic properties, the capillary model [63] was combined with a water retention curve for analysis. The capillary model considers that the matric potential $S$ is mainly the result of capillary forces acting on circular capillaries of a certain range of pore sizes. In the model, $\sigma$ is used to denote the water surface tension coefficient ($7.5 \times 10^{-4}$ N/cm at room temperature), $r_0$ denotes the capillary radius, and $D$ denotes the capillary diameter (i.e., equivalent pore diameter, $D = 2r_0$). The relationship between the equivalent pore diameter and the matric potential is $D = 4\sigma/S$. When the matric potential is $S_1$, the corresponding equivalent pore diameter is $D_1$. Only in the pore diameter less than $D_1$ are capillary pores filled with water, and the water content is $\theta_1$. When the matric potential is $S_2$ ($S_1 < S_2$), $D_2$, $\theta_2$ are obtained in the same way. The ratio of the pore volume occupied by pores with an equivalent pore size between $D_2$ and $D_1$ to the total volume of substrate pores is called the equivalent pore volume ratio ($\theta_1$–$\theta_2$). Based on the above theory, equivalent pore volume ratios of PAS between $0$––$10$ kPa, $-10$––$100$ kPa, $-100$––$1000$ kPa were calculated (Table 5). As can be seen from Table 5, compared to vegetated PAS, pure PAS has greater volume ratios of pores, with diameters > 0.03 mm and between 0.003–0.03 mm, but lower ratios of pores with diameters <0.003 mm. According to the agronomic criteria, pores larger than 0.03 mm in diameter tend to act as macropores for water permeable and aeration, and water in pores between 0.003–0.03 mm are most easily accessible to plants [35]. The differences in pore structure between vegetated PAS and pure PAS reflect that the presence of roots in PAS can

effectively block its macropores. Further analysis combined with the root characteristics (Table 2) reveal that the volume ratio of pores > 0.03 mm in diameter is linearly correlated (correlation coefficient $r$ = 0.92) with the root volume density of 0.2–0.4 mm roots [39]. This indicates that for vegetated PAS, although shallow-rooted *Sedums* with a large portion of fine roots often lead to a reduction of macropores, the presence of 0.2–0.4 mm roots can effectively offset the reduction. The root characteristics, in turn, are associated with interactions between plant species, substrate type, and substrate depth [60]. The relatively high (28.48–30.63%) macropores are present in 6 cm-PAS-*SS*, 10 cm-PAS-*SS*, and 6 cm-PAS-*SL* (Table 5). It is noted that a deeper PAS does not promote the development of 0.2–0.4 mm roots (Table 2); on the contrary, it will foster a root system resulting in macropore blockage and $K_s$ reduction [60].

**Table 5.** Equivalent pore volume ratio of PAS.

| Name | Range of Equivalent Pore Sizes (Corresponding Matric Potentials) | | |
| --- | --- | --- | --- |
| | >0.03 mm (0–−10 kPa) | 0.003–0.03 mm (−10–−100 kPa) | 0.0003–0.003 mm (−100–−1000 kPa) |
| 6 cm-PAS-*SS* | 28.48% | 5.34% | 0.42% |
| 10 cm-PAS-*SS* | 30.60% | 6.34% | 0.61% |
| 14 cm-PAS-*SS* | 0.99% | 12.47% | 12.30% |
| 6 cm-PAS-*SL* | 30.63% | 6.12% | 0.60% |
| 10 cm-PAS-*SL* | 4.18% | 18.80% | 6.92% |
| 14 cm-PAS-*SL* | 4.30% | 21.05% | 5.42% |
| pure PAS | 41.63% | 48.41% | 0.11% |

*3.3. Preferential Flow and Influential Factors*

3.3.1. Perlite-Based Substrate (PAS)

With the substrate hydraulic parameter (Table 4) in the Hydrus-1D model, various simulation conditions (Table 6) were set up to systematically investigate the effects of plant species, substrate depth, rainfall intensity, and initial water content on the preferential outflow in PAS. The results of the skewness coefficient (*S*), preferential outflow, and PFI under each simulation condition are shown in Table 6. All of the *S* values are not zero, suggesting that the occurrence of preferential flow is prevalent. The preferential outflow all exceed 2.49 cm, and the PFI ranges from 33.00% to 100.00%.

Based on the simulation results (Table 6), multi-factor ANOVA was performed, and the results are shown in Table 7. It can be seen that for PAS, rainfall intensity, plant species, substrate depth, and the interaction of plant species and substrate depth, all had significant effects on the preferential outflow and PFI, while the initial water content had no significant effect on both. Therefore, simulation conditions with the initial water content of WHC were excluded from the following analysis, which focuses on rainfall intensity, plant species, and substrate depth for 27 simulation conditions only.

Rainfall intensity: It can be seen from *F* values in Table 7 that rainfall intensity has the greatest effect on preferential outflow (*F* = 268.98), and a correlation analysis for the 27 simulation conditions shows a positive (correlation coefficient $r$ = 0.83) linear relationship between the two. When the rainfall intensity varied with fixed other factors (plant species, substrate depth, etc.), among all the simulation conditions (Table 8), preferential outflow produced from 10 a-rainfall was greater than that from 5 a-rainfall, which in turn was greater than that from 2 a-rainfall (Table 6). However, the rainfall intensity influenced PFI to a lesser extent (*F* = 8.175). The positive (correlation coefficient $r$ = 0. 78) linear correlation between PFI and rainfall intensity also exists for the 27 simulation conditions. It is noted that in Table 8, with rainfall intensity varying, high mean PFI ($\geq$67.19%), but low $C_v$ ($\leq$2.97%) are present in non-vegetated PAS, 6 cm- and 10 cm-PAS with *SS*, and 6 cm-PAS with *SL*. Table 5 reveals that these plant–substrate combinations have high portions of macropores (28.48–41.63%), comprising pore networks favoring preferential flow devel-

opment [34]. Therefore, preferential flow development in these plant–substrates is mainly influenced by internal pore structure and less correlated with rainfall intensity, resulting in high mean PFI, but low $C_v$ values. In contrast, for 14 cm-PAS with *SS*, and 10 and 14 cm-PAS with *SL*, small portions of macropores (0.99–4.30%, Table 5) provide few preferential paths, and therefore, the related preferential flow development can be influenced by both internal pore structure and rainfall intensity. Correspondingly, the associated mean PFI are 57.51–82.50%, 36.04–51.06%, and 33.00–65.68%, respectively, and the $C_v$ values are 17.77%, 17.47%, and 38.63%, all showing considerable degrees of variability (Table 8).

Plant species: The effect of plant species on preferential outflow (*F* = 118.54, Table 7) is second only to rainfall intensity (*F* = 268.98, Table 7). When plant species varied with fixed other factors (substrate depth, rainfall intensity, etc.), among all the simulation conditions (Table 9), preferential outflow $C_v$ values from 6 cm-PAS subject to various rainfalls were less than 10%, while those from 10 cm-PAS and 14 cm-PAS subject to various rainfalls were greater than 20%. Considering the secondary importance of plant species for preferential outflow (Table 7), the change in preferential outflow $C_v$ can be attributed to plant species, and the effect of plant species on preferential outflow becomes more prominent for deeper substrates. In addition, for any three simulations of varying plant species, but fixed other factors (Table 9), non-vegetated PAS had the largest preferential outflow (Table 6). This may be due to a high volume ratio of macropores (>0.03 mm) in the non-vegetated PAS (41.63%, Table 5), which was 1.36–42.05 times larger than that in the vegetated PAS, and since macropores are potential preferential flow paths, eventually the largest preferential outflow occurred in non-vegetated PAS. Table 7 also shows that plant species have the greatest effect on PFI (*F* = 84.98). Similar to preferential outflow, based on changes in PFI $C_v$ values for simulations of varying plant species, it can be concluded that the effect of plant species on PFI also becomes more prominent for deeper substrates. As plants make changes to the pore structures of PAS (Table 5), further analysis for the 27 simulation conditions shows a positive (correlation coefficient *r* = 0.92) linear correlation between macropore volume ratio and PFI. Since the macropore volume ratio is also significantly and positively correlated with the root volume density of 0.2–0.4 mm roots (Section 3.2.2), it indicates that *Sedum* roots of 0.2–0.4 mm diameter promote the development of preferential flow.

Substrate depth: The effect of substrate depth on the preferential outflow is the smallest (*F* = 31.66, Table 7). When the substrate depth varied with fixed other factors (plant species, rainfall intensity, etc.), among all the simulation conditions (Table 10), preferential outflow $C_v$ values from non-vegetated PAS subject to various rainfalls were less than 1%, followed by less than 20% from PAS with *SS*, and greater than 27% from PAS with *SL* (Table 10). This indicates that PAS with *SL* is more influenced by substrate depth in terms of preferential outflow, compared to non-vegetated PAS and PAS with *SS*. In addition, for any three simulations of varying substrate depth, but fixed other factors (Table 10), 6 cm-vegetated PAS had the largest preferential outflow (Table 6). Similar to preferential outflow, the effect of substrate depth on PFI was the smallest (*F* = 23.94, Table 7), and based on PFI $C_v$ changes, it is concluded that PAS with *SL* is more influenced by substrate depth in terms of PFI, compared to non-vegetated PAS and PAS with *SS*. Likewise, for any three simulations of varying substrate depth, but fixed other factors (Table 10), 6 cm-vegetated PAS had the largest PFI (Table 6). The 6 cm depth of vegetated PAS is associated with high root volume densities of 0.2–0.4 mm roots (0.33 mm$^3$/cm$^3$ for 6 cm-PAS-*SS* and 0.31 mm$^3$/cm$^3$ for 6 cm-PAS-*SL*, Table 2) that can play positive roles for preferential flow development.

**Table 6.** Simulation results of PAS under different simulation conditions.

| Simulation No. | Plant Species | Substrate Depth/(cm) | Rainfall Intensity/(a) | Initial Water Content/(%) | S | Preferential Outflow/(cm) | PFI/(%) |
|---|---|---|---|---|---|---|---|
| 1 | | | 2 | WHC | −0.35 | 6.38 | 84.29 |
| 2 | | | | MDC | −0.35 | 6.38 | 84.29 |
| 3 | | 6 | 5 | WHC | −0.06 | 9.01 | 83.43 |
| 4 | | | | MDC | −0.06 | 9.01 | 83.43 |
| 5 | | | 10 | WHC | −0.31 | 11.05 | 83.65 |
| 6 | | | | MDC | −0.31 | 11.05 | 83.65 |
| 7 | | | 2 | WHC | 0.64 | 5.08 | 67.19 |
| 8 | | | | MDC | 0.64 | 5.08 | 67.19 |
| 9 | *Sedum sarmentosum* | 10 | 5 | WHC | 0.04 | 7.59 | 70.32 |
| 10 | | | | MDC | 0.04 | 7.59 | 70.32 |
| 11 | | | 10 | WHC | 0.27 | 9.37 | 70.77 |
| 12 | | | | MDC | 0.04 | 9.37 | 70.77 |
| 13 | | | 2 | WHC | 0.34 | 4.35 | 57.51 |
| 14 | | | | MDC | 0.34 | 4.35 | 57.51 |
| 15 | | 14 | 5 | WHC | 0.12 | 7.71 | 71.43 |
| 16 | | | | MDC | 0.12 | 7.71 | 71.43 |
| 17 | | | 10 | WHC | 0.56 | 10.89 | 82.50 |
| 18 | | | | MDC | 0.56 | 10.89 | 82.50 |
| 19 | | | 2 | WHC | 0.37 | 6.73 | 89.00 |
| 20 | | | | MDC | 0.37 | 6.73 | 89.00 |
| 21 | | 6 | 5 | WHC | −0.27 | 9.56 | 88.55 |
| 22 | | | | MDC | −0.27 | 9.56 | 88.55 |
| 23 | | | 10 | WHC | −0.39 | 11.58 | 87.73 |
| 24 | | | | MDC | 0.39 | 11.58 | 87.73 |
| 25 | | | 2 | WHC | 0.88 | 2.72 | 36.04 |
| 26 | | | | MDC | 0.88 | 2.72 | 36.04 |
| 27 | *Sedum lineare* | 10 | 5 | WHC | 0.37 | 4.61 | 42.37 |
| 28 | | | | MDC | 0.37 | 4.61 | 42.37 |
| 29 | | | 10 | WHC | −0.52 | 6.74 | 51.06 |
| 30 | | | | MDC | −0.52 | 6.74 | 51.06 |
| 31 | | | 2 | WHC | 0.07 | 2.49 | 33.00 |
| 32 | | | | MDC | 0.07 | 2.49 | 33.00 |
| 33 | | 14 | 5 | WHC | 0.18 | 4.09 | 38.01 |
| 34 | | | | MDC | 0.18 | 4.09 | 38.01 |
| 35 | | | 10 | WHC | 0.22 | 8.66 | 65.68 |
| 36 | | | | MDC | 0.22 | 8.66 | 65.68 |
| 37 | | | 2 | WHC | 0.09 | 7.59 | 99.97 |
| 38 | | | | MDC | 0.09 | 7.59 | 99.97 |
| 39 | | 6 | 5 | WHC | 0.10 | 10.85 | 100.0 |
| 40 | | | | MDC | 0.10 | 10.85 | 100.0 |
| 41 | | | 10 | WHC | 0.11 | 13.21 | 99.92 |
| 42 | | | | MDC | 0.11 | 13.21 | 99.92 |
| 43 | | | 2 | WHC | 0.01 | 7.56 | 99.89 |
| 44 | | | | MDC | 0.01 | 7.56 | 99.89 |
| 45 | No-plants | 10 | 5 | WHC | 0.59 | 10.73 | 99.91 |
| 46 | | | | MDC | 0.59 | 10.73 | 99.91 |
| 47 | | | 10 | WHC | 0.63 | 13.08 | 99.92 |
| 48 | | | | MDC | 0.63 | 13.08 | 99.92 |
| 49 | | | 2 | WHC | 0.18 | 7.59 | 99.87 |
| 50 | | | | MDC | 0.18 | 7.59 | 99.87 |
| 51 | | 14 | 5 | WHC | 0.21 | 10.76 | 99.91 |
| 52 | | | | MDC | 0.21 | 10.76 | 99.91 |
| 53 | | | 10 | WHC | 0.46 | 13.19 | 99.77 |
| 54 | | | | MDC | 0.46 | 13.19 | 99.77 |

**Table 7.** Multi-factor ANOVA results for preferential outflow and PFI of PAS.

| Sources of Variance | *F* Values | |
| --- | --- | --- |
| | Preferential Outflow | PFI |
| Plant species | 118.54 ** | 84.98 ** |
| Substrate depth | 31.66 ** | 23.94 ** |
| Plant species × Substrate depth | 13.96 ** | 10.55 ** |
| Rainfall intensity | 268.98 ** | 8.175 * |
| Initial water content | 0.00 | 0.00 |

Note: * ($p < 0.05$) reached a significant level and ** ($p < 0.01$) reached a highly significant level.

**Table 8.** Mean values and $C_v$ of PAS preferential outflow and PFI for different rainfall intensities.

| Simulation No. (Rainfall Intensity Varying) | Fixed Variables: Plant Species, Substrate Depth, Initial Water Content | Preferential Outflow | | PFI | |
| --- | --- | --- | --- | --- | --- |
| | | Mean Values /(cm) | $C_v$ /(%) | Mean Values /(cm) | $C_v$ /(%) |
| 2, 4, 6 | *SS*, 6 cm, MDC | 8.81 | 26.56 | 83.79 | 0.53 |
| 20, 22, 24 | *SL*, 6 cm, MDC | 9.29 | 26.22 | 88.43 | 0.73 |
| 38, 40, 42 | No-plants, 6 cm, MDC | 10.55 | 26.75 | 99.96 | 0.04 |
| 8, 10, 12 | *SS*, 10 cm, MDC | 7.35 | 29.30 | 69.36 | 2.97 |
| 26, 28, 30 | *SL*, 10 cm, MDC | 4.69 | 42.88 | 43.16 | 17.47 |
| 44, 46, 48 | No-plants, 10 cm, MDC | 10.46 | 26.49 | 99.91 | 0.02 |
| 14, 16, 18 | *SS*, 14 cm, MDC | 7.65 | 42.75 | 70.48 | 17.77 |
| 32, 34, 36 | *SL*, 14 cm, MDC | 3.20 | 63.03 | 45.56 | 38.63 |
| 50, 52, 54 | No-plants, 14 cm, MDC | 10.51 | 26.71 | 99.85 | 0.07 |

**Table 9.** Mean values and $C_v$ of PAS preferential outflow and PFI for different plant species.

| Simulation No. (Plant Species Varying) | Fixed Variables: Substrate Depth, Rainfall Intensity, Initial Water Content | Preferential Outflow | | PFI | |
| --- | --- | --- | --- | --- | --- |
| | | Mean Values /(cm) | $C_v$ /(%) | Mean Values /(cm) | $C_v$ /(%) |
| 2, 20, 38 | 6 cm, 2 a, MDC | 6.90 | 9.02 | 91.09 | 8.83 |
| 4, 22, 40 | 6 cm, 5 a, MDC | 9.81 | 9.63 | 90.66 | 9.36 |
| 6, 24, 42 | 6 cm, 10 a, MDC | 11.95 | 9.42 | 90.43 | 9.36 |
| 8, 26, 44 | 10 cm, 2 a, MDC | 5.12 | 47.26 | 67.71 | 47.16 |
| 10, 28, 46 | 10 cm, 5 a, MDC | 7.64 | 40.04 | 70.87 | 40.60 |
| 12, 30, 48 | 10 cm, 10 a, MDC | 9.73 | 32.74 | 73.92 | 33.26 |
| 14, 32, 50 | 14 cm, 2 a, MDC | 4.81 | 53.66 | 63.46 | 53.31 |
| 16, 34, 52 | 14 cm, 5 a, MDC | 7.52 | 44.40 | 69.78 | 44.40 |
| 18, 36, 54 | 14 cm, 10 a, MDC | 10.91 | 20.76 | 82.65 | 20.62 |

**Table 10.** Mean values and $C_v$ of PAS preferential outflow and PFI for different substrate depths.

| Simulation No. (Substrate Depth Varying) | Fixed Variables: Plant Species, Rainfall Intensity, Initial Water Content | Preferential Outflow | | PFI | |
| --- | --- | --- | --- | --- | --- |
| | | Mean Values /(cm) | $C_v$ /(%) | Mean Values /(cm) | $C_v$ /(%) |
| 2, 8, 14 | *SS*, 2 a, MDC | 5.08 | 20.01 | 67.19 | 19.98 |
| 4, 10, 16 | *SS*, 5 a, MDC | 8.10 | 9.72 | 75.06 | 9.69 |
| 6, 12, 18 | *SS*, 10 a, MDC | 10.44 | 8.88 | 78.97 | 9.03 |
| 20, 26, 32 | *SL*, 2 a, MDC | 3.98 | 59.91 | 52.68 | 59.78 |
| 22, 28, 34 | *SL*, 5 a, MDC | 6.09 | 49.60 | 56.31 | 49.73 |
| 24, 30, 36 | *SL*, 10 a, MDC | 8.99 | 27.10 | 68.16 | 27.08 |
| 38, 44, 50 | No-plants, 2 a, MDC | 7.58 | 0.23 | 99.91 | 0.05 |
| 40, 46, 52 | No-plants, 5 a, MDC | 10.78 | 0.58 | 99.94 | 0.05 |
| 42, 48, 54 | No-plants, 10 a, MDC | 13.16 | 0.53 | 99.87 | 0.09 |

### 3.3.2. Vermiculite-Based Substrate (VAS)

Since the hydraulic parameters of varying VAS remain unchanged or appear comparable (Table 4), minor changes in the pore structures of VAS due to *Sedum* root system are expected. Therefore, the influential factors considered for VAS were substrate depth, rainfall intensity, and initial water content, for which simulation conditions involving no-plants only were set up in Table 11. The resulted skewness coefficient *S*, preferential outflow, and PFI are also listed in Table 11. None of the *S* values are zero, which indicates the prevalence of preferential flow occurrence in VAS. The minimum values of preferential outflow and PFI are 7.04 cm and 93.06%, respectively.

**Table 11.** Simulation results of VAS under different simulation conditions.

| Simulation Conditions | Substrate Depth /(cm) | Rainfall Intensity /(a) | Initial Water Content /(%) | *S* | Preferential Outflow /(cm) | PFI /(%) |
|---|---|---|---|---|---|---|
| 1 | | 2 | WHC | 0.05 | 7.42 | 98.27 |
| 2 | | | MDC | 0.05 | 7.42 | 98.27 |
| 3 | 6 | 5 | WHC | 0.06 | 10.62 | 98.33 |
| 4 | | | MDC | 0.06 | 10.62 | 98.33 |
| 5 | | 10 | WHC | 0.30 | 13.01 | 98.26 |
| 6 | | | MDC | 0.30 | 13.01 | 98.26 |
| 7 | | 2 | WHC | 0.05 | 7.24 | 95.81 |
| 8 | | | MDC | 0.05 | 7.24 | 95.81 |
| 9 | 10 | 5 | WHC | 0.23 | 10.36 | 95.84 |
| 10 | | | MDC | 0.23 | 10.36 | 95.84 |
| 11 | | 10 | WHC | 0.01 | 12.67 | 95.84 |
| 12 | | | MDC | 0.01 | 12.67 | 95.84 |
| 13 | | 2 | WHC | −0.61 | 7.04 | 93.09 |
| 14 | | | MDC | −0.61 | 7.04 | 93.09 |
| 15 | 14 | 5 | WHC | 0.17 | 10.05 | 93.06 |
| 16 | | | MDC | 0.17 | 10.05 | 93.06 |
| 17 | | 10 | WHC | 0.26 | 12.31 | 93.12 |
| 18 | | | MDC | 0.26 | 12.31 | 93.12 |

Multi-factor ANOVA was also performed based on the above simulation results (Table 12). It can be seen that for VAS, both rainfall intensity and substrate depth had significant effects on preferential outflow, and substrate depth had a significant effect on PFI. However, initial water content had no significant effect on preferential outflow and PFI. Therefore, simulation conditions with the initial water content of WHC were excluded from the following analysis, which focuses on rainfall intensity and substrate depth for nine simulation conditions only.

**Table 12.** Multi-factor ANOVA results for preferential outflow and PFI of VAS.

| Sources of Variance | *F* Values | |
|---|---|---|
| | Preferential Outflow | PFI |
| Substrate depth | 104.095 ** | 50,845.585 ** |
| Rainfall intensity | 10,207.964 ** | 1.098 |
| Initial water content | 0.000 | 0.000 |

Note: ** ($p < 0.01$) reached a highly significant level.

Rainfall intensity: The *F* value of 10,207.964 (Table 12) clearly shows the dominant role of rainfall intensity on preferential outflow, and a correlation analysis for the nine simulation conditions also shows a positive (correlation coefficient $r = 0.98$) linear relationship between the two. The larger the rainfall intensity, the greater the preferential outflow produced. In contrast, rainfall intensity had no significant effect on PFI (Table 12). When the nine simulation conditions of varying rainfall intensity were grouped by the substrate depth

(Table 13), the resulting mean PFI values are high ($\geq$93.06%), while the $C_v$ values are extremely low ($\leq$0.1%).

**Table 13.** Mean values and $C_v$ of VAS preferential outflow and PFI for different rainfall intensities.

| Simulation No. (Rainfall Intensity Varying) | Fixed Variables: Substrate Depth, Plant Species, Initial Water Content | Preferential Outflow | | PFI | |
|---|---|---|---|---|---|
| | | Mean Values /(cm) | $C_v$ /(%) | Mean Values /(cm) | $C_v$ /(%) |
| 2, 4, 6 | 6 cm, no-plants, MDC | 10.35 | 27.09 | 98.29 | 0.04 |
| 8, 10, 12 | 10 cm, no-plants, MDC | 10.09 | 26.98 | 95.83 | 0.02 |
| 14, 16, 18 | 14 cm, no-plants, MDC | 9.80 | 26.97 | 93.09 | 0.03 |

Substrate depth: In addition to the rainfall intensity, VAS depth also had an effect on preferential outflow (Table 12, *F* = 104.095). When the nine simulation conditions of varying substrate depth were grouped by the rainfall intensity (Table 14), it was noted that in a rainfall event, preferential outflow from 6 cm-VAS was the largest and from 14 cm-VAS was the smallest (Table 11). The $C_v$ values of preferential outflow were small (<3%, Table 14). The PFI was only influenced by the VAS depth (Table 12), and its variation pattern, along with different substrate depths, was consistent with that of the preferential outflow (e.g., 6 cm-VAS had the largest PFI, and the $C_v$ values had limited variations (Table 14)).

**Table 14.** Mean values and $C_v$ of VAS preferential outflow and PFI for different substrate depths.

| Simulation No. (Substrate Depth Varying) | Fixed Variables: Rainfall Intensity, Plant Species, Initial Water Content | Preferential Out-flow | | PFI | |
|---|---|---|---|---|---|
| | | Mean Values /(cm) | $C_v$ /(%) | Mean Values /(cm) | $C_v$ /(%) |
| 2, 8, 14 | 2 a, no-plants, MDC | 7.24 | 2.62 | 95.72 | 2.71 |
| 4, 10, 16 | 5 a, no-plants, MDC | 10.34 | 2.76 | 95.74 | 2.75 |
| 6, 12, 18 | 10 a, no-plants, MDC | 12.66 | 2.76 | 95.74 | 2.69 |

Considering the negative role of preferential flow on green roof stormwater performance [23,64], the degree of preferential flow should be minimized as much as possible in green roofs. The above analysis shows that preferential flow development in PAS and VAS are controlled by different factors. For PAS, rainfall intensity, plant species, and substrate depth all had significant effects on PFI (Table 7), while for VAS, only substrate depth played a role on PFI. After reviewing Tables 8 and 13, it is known that 10 cm-PAS-*SL* has the lowest mean PFI of 43.16%, regardless of rainfall intensity, and all VAS have large mean PFI ranging from 93.09% to 98.29%. Therefore, for the preferential flow control purpose, a combination of 10 cm-PAS-*SL* may be recommended for the plant–substrate design in green roofs. It should be noted that this recommendation is made based on green roof performance for individual large rainfalls. Future research focusing on improving green roof performance for both large and small rainfalls over a long period may come up with a better plant–substrate design recommendation.

## 4. Conclusions

In order to investigate the law of preferential outflow in various green roof plant–substrate combinations, two *Sedum* plants (namely *Sedum sarmentosum* and *Sedum lineare*) were planted in two artificial substrates (namely, PAS and VAS) at three different depths, and pure artificial substrates were also set as controls. Thereafter, indoor solute breakthrough experiments and water flow and solute transport simulations in Hydrus-1D were conducted. The in-door experimental results showed that the skewness coefficients of all solute breakthrough curves were non-zero, indicating preferential flow generally occurred in green roof plant–substrate combinations. The hydraulic parameters of different substrates were obtained from the inverse modeling in Hydrus-1D. The correlation coefficients between the modeled and measured values of the cumulative outflow for PAS and VAS were in the range of 0.998–0.999, and the Nash–Sutcliffe efficiency coefficients were in the

range of 0.741–0.997. For PAS, the different hydraulic parameters of the vegetated PAS at different depths were due to the differences in root-induced pore structure. In contrast, hydraulic parameters from different VAS can be viewed as the same. According to the forward modeling results in Hydrus-1D, it is concluded that for PAS, rainfall intensity, plant species, substrate depth, and the interaction of plant species and substrate depth all had significant effects on the preferential outflow and PFI, while the initial water content had no significant effect on both. For VAS, both rainfall intensity and substrate depth had significant effects on the preferential outflow, and substrate depth also had a significant effect on PFI. Likewise, initial water content had no significant effect on VAS preferential outflow and PFI. The 10 cm-PAS with *S. lineare* may be recommended for preferential flow control purposes. Further research considering both preferential flow control and stormwater retention improvement for green roof design is needed.

**Author Contributions:** Investigation, X.C.; Formal analysis, X.C. and R.L.; Methodology, R.L. and D.L.; Writing—Original Draft Preparation, X.C. and R.L.; Writing—Review & Editing, R.L. and X.X. All authors have read and agreed to the published version of the manuscript.

**Funding:** The research was funded by the National Natural Science Foundation of China, grant number [51909081].

**Data Availability Statement:** The data presented in this study are available from the first author upon reasonable request.

**Conflicts of Interest:** The authors declare no conflict of interest.

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
