# Peer review of "Analysis of Preferential Flow in Artificial Substrates with Sedum Roots for Green Roofs: Experiments and Modeling"

_water, doi:10.3390/w15050914_

Round 1

Reviewer 1 Report

The authors investigated the Investigation Analysis of preferential flow in artificial substrates with Sedum roots for green roofs: experiments and modeling. The authors designed the work systematic way with performing some valuable experimental works accordingly. It is also necessary to critically evaluate new data and do not make hasty conclusions which may lead to misinterpretations.

However, several points are important to be addressed before going to possible publication in this high-quality journal. Also, the authors need to address all points in the revision stage for broad range readers understanding.

the materials and methods should be described in detail.

Comparative study should be included.

Some Future prospects should be included in the conclusion part.

The novelty of the work should be established.

Many grammar errors could be found in the paper, please check and deeply revise them. ‎Also, the English language of the paper should be improved. ‎

Quantitative information should be provided in the abstract.

Reviewer 2 Report

The introduction, cited references, research design, and methods are sufficient, relevant, appropriate, and adequately described, respectively. The results and conclusions can be improved. Some figures, tables, and texts can be improved as well. The detailed comments are as follows.

1. L44-46. The corresponding sentence “The detention … runoff [6].” can be improved.

2. The authors described in the introduction as follows. “The purposes of this paper are (1) to detect the occurrence of preferential flow in various plant-substrate combinations by indoor solute breakthrough experiments, (2) to characterize the substrate hydraulic properties of each combination, (3) to analyse the effects of plant type, substrate depth, rainfall intensity, and initial water content on the preferential flow development in plant-substrate combinations.” Need to check if the conclusion includes all the answers of the purposes.

3. Need to explain why the 103-day artificial cultivation was scheduled.

4. The number of the total combination (i.e., 12) including substrate depth, substrate type, and plant type should be mentioned when the authors describe Table 2.

5. L131-132. The authors selected WHC and MDC for initial water content and cited the reference [37]. Need to specify whether the selection was made according to the reference [37] with a proper reason; otherwise, need to specify the connection between the reference [37] and the selection.

6. L132-135. The descriptions were somewhat confused because it is hard to obtain the final number of 54 sets. Need to describe those in a clearer way. Besides, it is necessary to explain what the three lines (solid, dashed, and dotted) represent in Fig. 1.

7. L184-186. It is better use mathematical notations instead of using “S is negative,” “S is positive,” “S is 0,” and “S is not 0.” For example, “S < 0”, “S > 0”, “S = 0”, etc.

8. L197-198. The m and f should be written in italic.

9. L208-L220, and other relevant expressions in the text. The minus sign in the superscripts should be a longer. For example, cm-3 should be cm–3.

10. L228-229. The values of alpha, beta, and gamma were determined according to the reference [47]. Need to support the reasons to using the values with more specific ways.

11. L250. Use the mathematical notation “–infinity notation (in equation) to 1” instead using “negative infinity to 1.”

12. In the captions of Table 8, Table 9, Table 10, Table 13, Table 14. The period after PFI should be deleted.

13. In the caption of Table 5, need to use a longer minus sign in “-10 kPa” “-100 kPa”, and “-1000 kPa.”

14. L358-359. The minus sign should be a longer.

15. In the text, Figure 16 is not mentioned. The figure should be mentioned in the relevant sentence. Besides, in horizontal axis of Figure 16, i.e., “matric potential (-kPa)”, need to specify what the minus sign stands for.

16. Figure 4 can be improved as follows. First, “modelled outflow” and “observed outflow” can be simplified to “modelled” and “observed”. Second, the horizontal axis “time (min) can be only shown in first three figures (i.e., PAS1, PAS2, and PAS 3), and then they can be omitted for the other figures.

17. Figure 5 can be improved as follows. First, “modelled concentration” and “observed concentration” can be simplified to “modelled” and “observed”. Second, the horizontal axis “time (min) can be only shown in first three figures (i.e., PAS1, PAS2, and PAS 3), and then they can be omitted for the other figures.

18. L310-L314. For the correlation coefficients R2 of Figure 4 and Figure 5, need to specify the details such as linear correlation etc. In particular, it is necessary to describe how the authors obtained the correlation coefficients R2 of Figure 5.

19. If possible, it is necessary to describe some tables by figures. That is because this paper has 14 tables and 6 figures.

20. In the Conclusion (and abstract), the sentence “The 10 cm-PAS with ….” is not likely fully supported by the descriptions above; hence, the authors need to consider rewriting the conclusion and abstract in clearer way. 

Reviewer 3 Report

I have two major areas of concern with this manuscript. 

The level of English in the manuscript is good, but nevertheless I feel that it would benefit from being read through carefully by an English language expert ( one matter to be addressed would to ensure that  ‘which’ is correctly used- a number of instances I think that ‘that’ would be appropriate . I will identify in my comments a number of places where I feel the wording or the grammar could be altered, but there may be more. The second aspect of the writing about which I am concerned is possibly more a matter for journal policy then criticism of the authors. I take the view that the Abstract is an important part of any paper, but it has a life of its own and is often read independently of the full paper. The Abstract is also restricted to a limited length but  I would consider that,  for example, any acronym introduced for the first time in the Abstract should have  the full name spelled out; if the first appearance is in the text it should also be spelled out full stop. 

 My other area of concern relates particularly  to botanical aspects of the study and the conduct of the studies. I recognise that the authors are not botanists  but  it is still important to get any discussion of issues correct. 

Lines 14 

 ..two  Sedum species --- not plants. 

Although further information about Sedum is not required in the Abstract, there could be further discussion in the text. Sedum  is a widely distributed genus in the northern hemisphere, and contains many species. Are the two species indigenous to China? Given the large number of  species why were these  two chosen for the study? Were the individual plants grown initially from seeds, or were they produced vegetatively and hence were clones. lacking genetic variation? 

On first mention of the species the names should include the nomenclatural authorities, or a standard flora cited as being the source of identification. After the first mention of Sedum 

In second mentions it does not to be in full -so Sedum sarmentosum, at first mention but subsequently S. sarmentosum. 

 line 15 

PAS and VAS 

On first mention, the acronyms should be fully spelled out. 

 Although use of artificial substrates has the advantage of uniformity for experimental reasons, has there been any study using natural substrates which demonstrates that the artificial substrates used provide for better growth.  

line 16 

Pure substrates without plants  were used as controls. This might be better wording 

Line 26 -7 

Substrate depth had a significant effect on VAS…. might be better  wording. 

Line 33  

 Use of green roofs is one of the important measures for stormwater management in the construction of sponge cities… 

 Might be preferable  wording. 

 While vegetated roofs are a feature of new buildings in many parts of the world the expression  ‘sponge cities’ is not common usage . An explanation of the concept  should be provided. 

Line 34 

 ….which usually… 

 As a matter of grammar/syntax it is not clear what ‘which’ refers to 

 …….. multiple functional layers,  among which the vegetation and substrate layers play important roles…. might be preferable wording 

 Line 37 

 ..  the leaves stems and branches intercept rainwater and the substrate layer  stores rainwater in its pore structure       might be better wording. 

 The discussion here is at a general level but presumably relates to experience in China. 

  I would ask a number of questions which could be discussed. 

 Most green roofs that I have observed are flat- but I have seen some vegetated sloping roofs. If the roof is sloping  then I would presume this might affect the hydraulic behaviour. Are any sloping roofs in China vegetated, and if so how would this be accommodated in the hydrological analysis? 

 The experimental substrate depths are relatively shallow. Depending upon the species of plant grown the eventual height of the vegetation might be considerable. Is the vegetation layer managed by pruning to limit the height. The experimental vegetation is established as a monoculture -in practice would this be colonised by other species over time? 

 In terms of overall ‘green’ strategy, a decision to install a green roof rules out that particular roof being used as a site solar power generation. Green roofs maybe important  for biodiversity-either through planting choices or as a result of natural colonisation by others species,   species rich assemblages of plants may develop, and provide valuable habitat for a diversity  of animals, of which most attention has been given to birds and insects. In a number of cities around the world there are now bee hives established on green roofs, and production of honey is an important benefit to building owners.  Are potential  biodiversity values of green roofs a consideration of any projects in China? Hydrologic aspects of green roofs for biodiversity would still be very important, but possibly other parameters would require to be considered.. 

 Green roofs are potentially heavy-and this may be a constraint limiting retrofitting of green roofs  to existing buildings,. For 10cm PAS + Sedum, which is the preferential option from this study, are there costs  of requiring a stronger roof than a conventional one  incurred in a new build? 

Line 43. 

.between rainwater input and runoff….. consumed by plants.. 

 A very small quantity of water taken up by plants is ‘consumed’ as a reactant in photosynthesis,mineral ions in the water would be incorporated into the plant- but the absolute amounts would be very small. 

 Most of the water is retained in cell vacuoles, and from the plant’s perspective is the means of generating and retaining turgor . 

 The amount of  water taken up by plants is related to the photosynthetic system. Sedum 

species are likely to be either facultative or obligatory CAM plants ( ie employ the Crassulacean Acid Metabolism path of photosynthesis)- was this a factor in choosing the species in the study? 

Line 44 

 Detention refers to the temporal…. 

 would be  preferable  wording. 

Line 57 -58 

 Green roofs, each with ….but the same substrate, in the Atlantic Canada.. 

 The substrate was the  same – but what was it? 

 Line 61 

grandiflora 

 refer to a flora as source of names or include  nomenclatural authorities 

 line 62 

 … but the same substrate in Shenzen 

 What was the substrate 

 Line 63 

 …the larger the plant root diameter.. 

 Does this mean the diameter of the root ball or the diameter of individual roots? 

..the weaker the rainfall retained.. 

Does this mean ‘ less rainfall retained?’ 

Line 68 

  …. relatively rare.        ‘at present’ is  not required. 

Line 72 

 heavy rainfall 

 Is there a difference in the quantity of rain and rainfall intensity? 

 Line 86 

 An artificial….. rather than ‘the’ 

 Line 98-99 

 This sentence needs to be re written 

 Line 116 –117 

 Growing in acrylic cylinders is ‘artificial’ – but were they also grown in a glasshouse? 

 Table 1 

 What were the plant available nutrients in the chicken  manure? Was the manure cinsistent in all of the studies. 

 Page 129-135 

 Where is ‘local’ 

 What was the time period for which local data were available? 

Line 141 

 ‘automatic weighing instrument’ 

 Does this mean ‘balance’ or ‘scales’? 

Line 144 

 Same intensity 

 What does this mean? 

Line363 

 ..less volume ratio.. 

 Should be this be ..lower ratios’.. 

Line 397 

 ..rest27 .. 

 rest of the 27? 

Line 422 

 Plant species ( or just species) 

Line 426 

 …rainfalls was…..while that from.. 

 Line 429 

 Plant species not type 

 Line 432 

 This may be due to a high…
